# DIFFERENTIABLE WEIGHTED FINITE-STATE TRANSDUCERS

## ABSTRACT

We introduce a framework for automatic differentiation with weighted finite-state transducers (WFSTs) allowing them to be used dynamically at training time. Through the separation of graphs from operations on graphs, this framework enables the exploration of new structured loss functions which in turn eases the encoding of prior knowledge into learning algorithms. We show how the framework can combine pruning and back-off in transition models with various sequence-level loss functions. We also show how to learn over the latent decomposition of phrases into word pieces. Finally, to demonstrate that WFSTs can be used in the interior of a deep neural network, we propose a convolutional WFST layer which maps lower-level representations to higher-level representations and can be used as a drop-in replacement for a traditional convolution. We validate these algorithms with experiments in handwriting recognition and speech recognition.

## 1 INTRODUCTION

Weighted finite-state transducers (WFSTs) are a commonly used tool in speech and language processing (Knight & May, 2009; Mohri et al., 2002). They are most frequently used to combine predictions from multiple already trained models. In speech recognition, for example, WFSTs are used to combine constraints from an acoustic-to-phoneme model, a lexicon mapping words to pronunciations, and a word-level language model. However, combining separately learned models using WFSTs only at inference time has several drawbacks, including the well-known problems of *exposure bias* (Ranzato et al., 2015) and *label bias* (Bottou, 1991; Lafferty et al., 2001).

Given that gradients may be computed for most WFST operations, using them only at the inference stage of a learning system is not a hard limitation. We speculate that this limitation is primarily due to practical considerations. Historically, hardware has not been sufficiently performant to make training with WFSTs tractable. Also, no implementation exists with the required operations which supports automatic differentiation in a high-level yet efficient manner.

We develop a framework for automatic differentiation through operations on WFSTs. We show the utility of this framework by leveraging it to design and experiment with existing and novel learning algorithms. Automata are a more convenient structure than tensors to encode prior knowledge into a learning algorithm. However, not training with them limits the extent to which this prior knowledge can be incorporated in a useful manner. A framework for differentiable WFSTs allows the model to jointly learn from training data as well as prior knowledge encoded in WFSTs. This enables the learning algorithm to incorporate such knowledge in the best possible way.

Use of WFSTs conveniently decomposes operations from data (*i.e.* graphs). For example, rather than hand-coding sequence-level loss functions such as Connectionist Temporal Classification (CTC) (Graves et al., 2006) or the Automatic Segmentation Criterion (ASG) (Collobert et al., 2016), we may specify the core assumptions of the criteria in graphs and compute the resulting loss with graph operations. This facilitates exploration in the space of such structured loss functions.

We show the utility of the differentiable WFST framework by designing and testing several algorithms. For example, bi-gram transitions may be added to CTC with a transition WFST. We scale transitions to large token set sizes by encoding pruning and back-off in the transition graph.

Word pieces are commonly used as the output of speech recognition and machine translation models (Chiu et al., 2018; Sennrich et al., 2016). The word piece decomposition for a word is learned

```
from gtn import *

def ASG(emissions, transitions, target):
  # Compute constrained and normalization graphs:
  A = intersect(intersect(target, transitions), emissions)
  Z = intersect(transitions, emissions)

  # Forward both graphs:
  A_score = forward_score(A)
  Z_score = forward_score(Z)

  # Compute loss:
  loss = negate(subtract(A_score, Z_score))

  # Clear previous gradients:
  emissions.zero_grad()
  transitions.zero_grad()

  # Compute gradients:
  backward(loss, retain_graph=False)
  return loss.item(), emissions.grad(), transitions.grad()
}
```

Figure 1: An example using the Python front-end of `gtn` to compute the ASG loss function and gradients. The inputs to the `ASG` function are all `gtn.Graph` objects.

with a task-independent model. Instead, we use WFSTs to marginalize over the latent word piece decomposition at training time. This lets the model learn decompositions salient to the task at hand.

Finally, we show that WFSTs may be used as layers themselves intermixed with tensor-based layers. We propose a convolutional WFST layer which maps lower-level representations to higher-level representations. The WFST convolution can be trained with the rest of the model and results in improved accuracy with fewer parameters and operations as compared to a traditional convolution.

In summary, our contributions are:

- A framework for automatic differentiation with WFSTs. The framework supports both C++ and Python front-ends and is available at `https://www.anonymized.com`.
- We show that the framework may be used to express both existing sequence-level loss functions and to design novel sequence-level loss functions.
- We propose a convolutional WFST layer which can be used in the interior of a deep neural network to map lower-level representations to higher-level representations.
- We demonstrate the effectiveness of using WFSTs in the manners described above with experiments in automatic speech and handwriting recognition.

## 2 RELATED WORK

A wealth of prior work exists using weighted finite-state automata in speech recognition, natural language processing, optical character recognition, and other applications (Breuel, 2008; Knight & May, 2009; Mohri, 1997; Mohri et al., 2008; Pereira et al., 1994). However, the use of WFSTs is limited mostly to the inference stage of a predictive system. For example, Kaldi, a commonly used toolkit for automatic speech recognition, uses WFSTs extensively, but in most cases for inference or to estimate the parameters of shallow models (Povey et al., 2011). In some cases, WFSTs are used statically to incorporate fixed lattices in discriminative sequence criteria (Kingsbury, 2009; Kingsbury et al., 2012; Su et al., 2013; Veselý et al., 2013).

Implementations of sequence criteria in end-to-end style training are typically hand-crafted with careful consideration for speed (Amodei et al., 2016; Collobert et al., 2019; Povey et al., 2016). The use of hand-crafted implementations reduces flexibility which limits research. In some cases, such as the fully differentiable beam search of Collobert et al. (2019), achieving the necessary computational efficiency with a WFST-based implementation may not yet be tractable. However, as a first step, we show that in many common cases we can have the expressiveness afforded by the differentiable WFST framework without paying an unacceptable penalty in execution time.

The ideas of learning with WFSTs (Eisner, 2002) and automatic differentiation through operations on graphs (Bottou et al., 1997) are not new. However, no simple and efficient frameworks exist. Especially related to this work, and inspiring the name of our framework, are the graph transformer networks of Bottou et al. (1997). Generalized graph transducers (Bottou et al., 1996) are more expressive than WFSTs, allowing arbitrary data as edge labels. When composing these graphs, one defines an edge matching function and a "transformer" to construct the resulting structure. While not this general, differentiable WFSTs nevertheless allow for a vast design space of interesting algorithms, and perhaps make a more pragmatic trade-off between flexibility and efficiency.

Highly efficient libraries for operations on WFSTs exist, notably OpenFST and its predecessor FSM (Allauzen et al., 2007; Mohri et al., 2000). We take inspiration from OpenFst in the interface and implementation of many of our functions. However, the design implications of operating on WFSTs with automatic differentiation are quite different than those of the use cases OpenFST has been optimized for. We also draw inspiration from libraries for automatic differentiation and deep learning (Collobert et al., 2011; Paszke et al., 2019; Pratap et al., 2019; Tokui et al., 2015).

Some of the algorithms we propose, with the goal of demonstrating the utility of the differentiable WFST library, are inspired by prior work. Prior work has explored pruning the set of allowed alignments with CTC, and in particular limiting the spacing between output tokens (Liu et al., 2018). Learning $n$-gram word decompositions with both differentiable (Liu et al., 2017) and non-differentiable (Chan et al., 2017) loss functions has also been explored.

## 3 DIFFERENTIABLE WEIGHTED FINITE-STATE TRANSDUCERS

A weighted finite-state acceptor $\mathcal{A}$ is a 6-tuple consisting of an alphabet $\Sigma$, a set of states $\mathbb{Q}$, start states $\mathbb{Q}_s$, accepting states $\mathbb{Q}_a$, a transition function $\pi(q, p)$ which maps elements of $\mathbb{Q} \times \Sigma$ to elements of $\mathbb{Q}$, and a weight function $\omega(q, p)$ which maps elements of $\mathbb{Q} \times \Sigma$ to $\mathbb{R}$. A weighted finite-state transducer $\mathcal{T}$ is a 7-tuple which augments an acceptor with an output alphabet $\Delta$. The transition function $\pi(q, p, r)$ and weight function $\omega(q, p, r)$ map elements of $\mathbb{Q} \times \Sigma \times \Delta$ to elements of $\mathbb{Q}$ and $\mathbb{R}$ respectively. In other words, each edge of a transducer connects two states and has an input label $p$, an output label $r$ and a weight $w \in \mathbb{R}$.

We denote an input by $\boldsymbol{p} = [p_1, \ldots, p_T]$ where each $p_i \in \Sigma$. An acceptor $\mathcal{A}$ *accepts* the input $\boldsymbol{p}$ if there exists a sequence of states $q_{i+1} = \pi(q_i, p_i) \in \mathbb{Q}$ such that $q_1 \in \mathbb{Q}_s$ and $q_{T+1} \in \mathbb{Q}_a$. With a slight abuse of notation, we let $\boldsymbol{p} \in \mathcal{A}$ denote that $\mathcal{A}$ accepts $\boldsymbol{p}$. The score of $\boldsymbol{p}$ is given by $s(\boldsymbol{p}) = \sum_{i=1}^T \omega(q_i, p_i)$. Let $\boldsymbol{r} = [r_1, \ldots, r_T]$ be a path with $r_i \in \Delta$. A transducer $\mathcal{T}$ *transduces* the input $\boldsymbol{p}$ to the output $\boldsymbol{r}$ (*i.e.* $(\boldsymbol{p}, \boldsymbol{r}) \in \mathcal{T}$) if there exists a sequence of states $q_{i+1} = \pi(q_i, p_i, r_i) \in \mathbb{Q}$ such that $q_1 \in \mathbb{Q}_s$ and $q_{T+1} \in \mathbb{Q}_a$. The score of the pair $(\boldsymbol{p}, \boldsymbol{r})$ is given by $s(\boldsymbol{p}, \boldsymbol{r}) = \sum_{i=1}^T \omega(q_i, p_i, r_i)$.

We restrict operations to the log and tropical semirings. In both cases the accumulation of weights along a path is with addition. In the log semiring the accumulation over path scores is with log-sum-exp which we denote by $\mathrm{logadd}_i \, s_i = \log \sum_i e^{s_i}$. In the tropical semiring, the accumulation over path scores is with the max, $\max_i s_i$. We allow $\epsilon$ transitions in both acceptors and transducers. An $\epsilon$ input (or output) on an edge means the edge can be traversed without consuming an input (or emitting an output). This allows inputs to map to outputs of differing length.

### 3.1 OPERATIONS

We briefly describe a subset of the most important operations implemented in our differentiable WFST framework. For a more detailed discussion of operations on WFSTs see *e.g.* Mohri (2009).

**Intersection** We denote by $\mathcal{B} = \mathcal{A}_1 \circ \mathcal{A}_2$ the intersection of two acceptors $\mathcal{A}_1$ and $\mathcal{A}_2$. The intersected graph $\mathcal{B}$ contains all paths which are accepted by both inputs. The score of any path in $\mathcal{B}$ is the sum of the scores of the two corresponding paths in $\mathcal{A}_1$ and $\mathcal{A}_2$.

**Composition** The same symbol denotes the composition of two transducers $\mathcal{U} = \mathcal{T}_1 \circ \mathcal{T}_2$. If $\mathcal{T}_1$ transduces $\boldsymbol{p}$ to $\boldsymbol{u}$ and $\mathcal{T}_2$ transduces $\boldsymbol{u}$ to $\boldsymbol{r}$ then $\mathcal{U}$ transduces $\boldsymbol{p}$ to $\boldsymbol{r}$. As in intersection, the score of the path in the composed graph is the sum of the path scores from the input graphs.

**Forward and Viterbi Score** The forward score of $\mathcal{T}$ is $\mathrm{logadd}_{(\boldsymbol{p}, \boldsymbol{r}) \in \mathcal{T}} \, s(\boldsymbol{p}, \boldsymbol{r})$. Similarly, the Viterbi score of $\mathcal{T}$ is $\max_{(\boldsymbol{p}, \boldsymbol{r}) \in \mathcal{T}} s(\boldsymbol{p}, \boldsymbol{r})$.

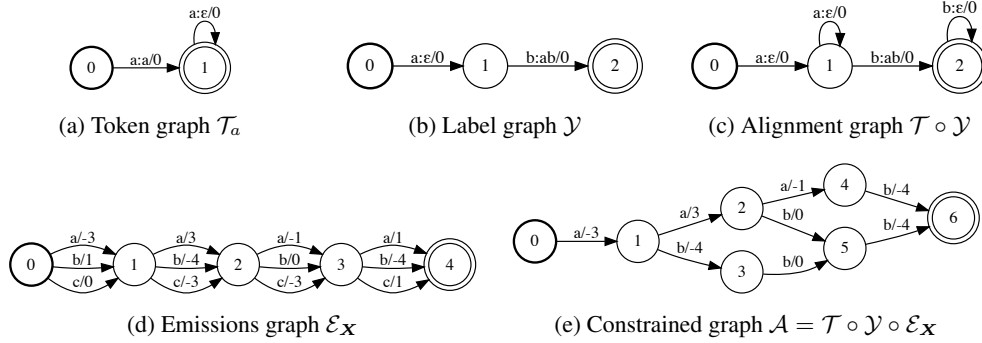

Figure 2: The graphs used to construct the ASG sequence criterion. The arc label "p:r/w" denotes an input label $p$, an output label $r$ and weight $w$. Graphs with just "p/w" are acceptors.

**Viterbi Path** The Viterbi path of $\mathcal{T}$ is given by $\arg\max_{(\boldsymbol{p},\boldsymbol{r})\in\mathcal{T}} s(\boldsymbol{p},\boldsymbol{r})$.

The forward score, Viterbi score, and the Viterbi path for an acceptor $\mathcal{A}$ are defined in the same way using just $\boldsymbol{p}$. In a directed acyclic graph these operations can be computed in time linear in the size of the graph with the well-known forward and Viterbi algorithms (Jurafsky, 2000; Rabiner, 1989).

We support standard rational operations including the union ($\mathcal{A} + \mathcal{B}$), concatenation ($\mathcal{AB}$), and the Kleene closure ($\mathcal{A}^*$) of WFSTs. To enable automatic differentiation through complete computations, we support slightly non-standard operations. We allow for the negation of all the arcs weights in a graph and the addition or subtraction of the arc weights of two identically structured graphs.

## 3.2 Automatic Differentiation

For the sake of automatic differentiation, every operation in section 3.1 accepts as inputs one or more graphs and returns a graph as output. For example, the forward score returns a "scalar" graph with a single arc between a start state and accepting state, with the score as the arc weight. The Viterbi path outputs the linear graph representing the best path $\boldsymbol{p}^*$ with $p_i^*$ as the arc labels. Jacobians of the output graph weights with respect to the weights of the input graphs may be computed for all of the operations described so far. Hence, the chain rule may be used to compute Jacobians or gradients of outputs with respect to the inputs of compositions of graph operations.

The WFST framework implements reverse-mode automatic differentiation following existing deep learning frameworks (Pratap et al., 2019; Paszke et al., 2019). Figure 1 gives an example Python implementation which computes gradients for the ASG criterion. In order to enable second-order differentiation, among other design considerations, the gradient with respect to a graph is also a graph. This adds little additional overhead as the two graphs share the underlying graph topology. The weights of the gradient graph are the gradients with respect to the weights of the graph.

## 4 Learning Algorithms

Let $\boldsymbol{X} = [\boldsymbol{x}_1, \dots \boldsymbol{x}_T] \in \mathbb{X}$ be a sequence of observations and $\boldsymbol{y} = [y_1, \dots, y_U] \in \mathbb{Y}$ a label. A sequence-level objective for the $(\boldsymbol{X}, \boldsymbol{y})$ pair can be specified with two weighted automata:

$$\log p(\boldsymbol{y} \mid \boldsymbol{X}) = \sum_{\boldsymbol{p}\in\mathcal{A}_{\boldsymbol{X},\boldsymbol{y}}} \mathrm{logadd}\, s(\boldsymbol{p}) - \sum_{\boldsymbol{p}\in\mathcal{Z}_{\boldsymbol{X}}} \mathrm{logadd}\, s(\boldsymbol{p}). \tag{1}$$

The target constrained graph, $\mathcal{A}_{\boldsymbol{X},\boldsymbol{y}}$, is constructed from the observation and target pair and the normalization graph, $\mathcal{Z}_{\boldsymbol{X}}$, is constructed only from the observation. We require $\mathcal{A}_{\boldsymbol{X},\boldsymbol{y}} \subseteq \mathcal{Z}_{\boldsymbol{X}}$.

The key ingredients to make equation 1 operational are 1) the structure of $\mathcal{A}$ and $\mathcal{Z}$ and 2) the source of the arc weights. The graph structures can be specified through functions on graphs. The initial arc weights come from the data itself or arbitrary learning algorithms (deep networks, $n$-gram language models, *etc.*) capable of providing useful scores on elements of $\mathbb{X}$ and $\mathbb{Y}$.

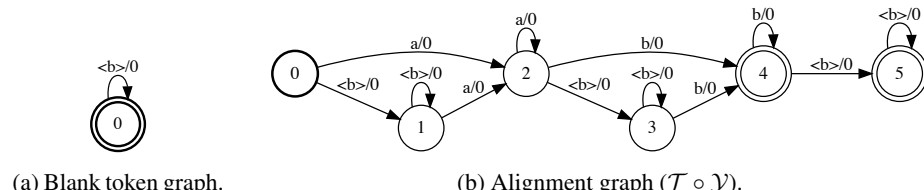

(a) Blank token graph.

(b) Alignment graph ($\mathcal{T} \circ \mathcal{Y}$).

Figure 3: The primary difference between ASG and CTC is the inclusion of the blank token graph (a) which allows for optional transitions on $<$b$>$ and results in the CTC alignment graph (b).

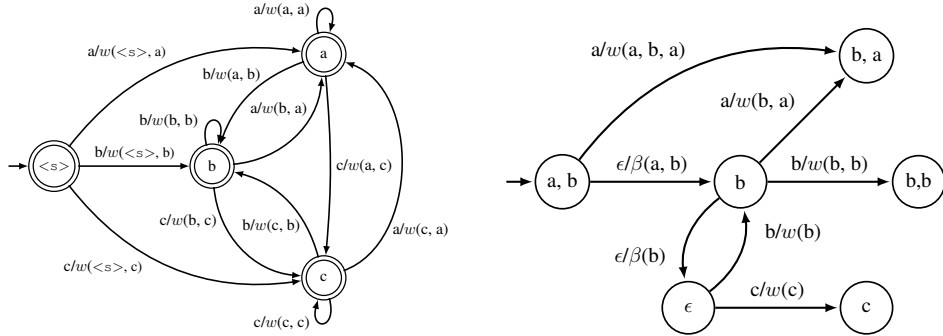

(a) A bigram graph for the token set {a,b,c}.

(b) An example of back-off in a trigram graph.

Figure 4: Transition graph examples. The $n$-gram score is $w$ and $\beta$ is the back-off weight.

The ASG (Collobert et al., 2016) and CTC (Graves et al., 2006) sequence criteria are commonly used in speech and handwriting recognition. See *e.g.* Hannun (2017) for background on the mechanics and motivation of CTC. To demonstrate the utility of our framework, we show how to construct these criteria from operations on simpler WFSTs following the formulation in equation 1.

The graphs $\mathcal{A}$ and $\mathcal{Z}$ used by the ASG loss function can be constructed from operations on simpler graphs. The ASG criterion assumes that emitting a new token requires consuming the token at least once, as in the graph, $\mathcal{T}_a$, of figure 2a. The full token graph is the closure of the union of the individual token graphs. Assuming an alphabet of {a, b, c} gives $\mathcal{T} = (\mathcal{T}_a + \mathcal{T}_b + \mathcal{T}_c)^*$. Composing $\mathcal{T}$ with the label graph $\mathcal{Y}$ (fig. 2b) of "ab" results in the alignment graph (fig. 2c). The alignment graph specifies the allowed correspondences between arbitrary length paths and the desired label. Intersecting the alignment graph with the emissions graph $\mathcal{E}_{\boldsymbol{X}}$ (fig. 2d) results in the constrained graph $\mathcal{A}$ (fig. 2e). The arc weights of $\mathcal{E}_{\boldsymbol{X}}$ depend on $\boldsymbol{X}$ and can be the output of a deep neural network. Here $\mathcal{Z}$ is simply the emissions graph. We may add bigram transitions by including a bigram transition graph, $\mathcal{B}$, as in figure 4a. In summary, the graph operations are:

$$\mathcal{A}_{\text{ASG}} = \mathcal{E} \circ (\mathcal{B} \circ ((\mathcal{T}_1 + \ldots + \mathcal{T}_C)^* \circ \mathcal{Y})) \quad \text{and} \quad \mathcal{Z}_{\text{ASG}} = \mathcal{E} \circ \mathcal{B}. \quad (2)$$

The primary differences between ASG and CTC are the use of bigram transitions in the former and the inclusion of a blank token in the latter. The blank token, $<$b$>$, is represented by the graph in figure 3a. Constructing CTC amounts to including the blank token graph when constructing the full token graph $\mathcal{T}$. The intersection $\mathcal{T} \circ \mathcal{Y}$ then results in the CTC alignment graph (fig. 3b). Note, this version of CTC does not force transitions on $<$b$>$ between repeats tokens. This requires remembering the previous state and hence is more involved (see Appendix A.1 for details).

A benefit of constructing sequence-level criteria by composing operations on simpler graphs is the access to a large design space of loss functions with which we can encode useful priors. For example we could construct a "spike" CTC, a "duration-limited" CTC, or an "equally spaced" CTC by substituting the appropriate token graphs into equation 2 (see Appendix A.2 for details).

## 4.1 TRANSITIONS

The ASG criterion was constructed with a bigram transition graph $\mathcal{B}$. We can use $\mathcal{B}$ to add transitions to CTC with the same operations in equation 2 but with the addition of the blank token.

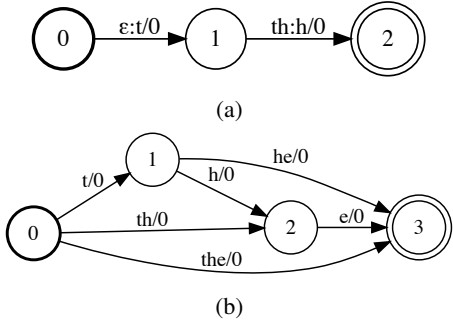 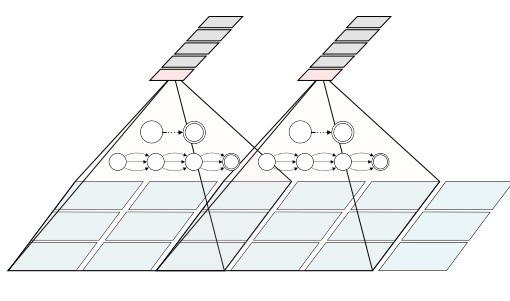

(a)

(b)

Figure 5: An individual sub-word-to-grapheme transducer (a) for the token "th" used to construct a lexicon $\mathcal{L}$ which is used to make the decomposition graph (b) for the label "the".

Figure 6: The convolutional transducer with a receptive field size of 3 and a stride of 2. Each output is the forward score of the composition of a kernel graph with a receptive field graph.

Dense $n$-gram graphs require $O(C^{n-1})$ states and $O(C^n)$ arcs for $C$ tokens, computationally intractable as $n$ and $C$ grow. The dense graph also suffers from sample efficiency problems as most transitions will not be observed in the training set. We can alleviate these issues with pruning and back-off (Katz, 1987) which can be encoded in a WFST (Mohri et al., 2008). Figure 4b demonstrates back-off for a trigram model. The trigram (a, b, a) exists so no back-off is needed. Neither the trigram (a, b, c) nor the bigram (b, c) exist, so the model backs off to the 0-gram state ($\epsilon$) and then transitions to the unigram state (c).

## 4.2 MARGINALIZED WORD PIECE DECOMPOSITIONS

Using word pieces has several benefits over grapheme outputs without the sample and computational efficiency issues, and lexical constraints of outputting words directly. However, the set of pieces and the decomposition for a word are learned in an unsupervised and task independent manner (Kudo, 2018; Sennrich et al., 2016). These are then fixed and used to train the task specific model. The word piece decomposition for a given phrase is not important, serving only as a stepping stone to more accurate models. This assumption can be made explicit by marginalizing over the set of decompositions for a target label while training the task specific model. This allows the model to select the decomposition(s) which are most salient to the task at hand.

Implementing marginalization over word piece decompositions usually requires non-trivial changes to carefully optimized implementations of existing sequence criterion. In the differentiable WFST framework this can be implemented in a plug-and-play fashion by incorporating a single lexicon graph. The lexicon transducer $\mathcal{L}$, which maps sequences of sub-word tokens to graphemes, is the closure of the union of the individual sub-word-to-grapheme graphs. A composition with the label graph, $\mathcal{L} \circ \mathcal{Y}$, gives the decomposition graph for the label $\boldsymbol{y}$. Figure 5 shows an example for the label "the". The decomposition graph can be used in place of the label graph to construct any down stream sequence criterion. For example, the constrained graph of equation 2 becomes:

$$\mathcal{A} = \mathcal{E} \circ (\mathcal{B} \circ ((\mathcal{T}_1 + \ldots + \mathcal{T}_C)^* \circ (\mathcal{L} \circ \mathcal{Y}))) \tag{3}$$

where $\mathcal{L} = (\mathcal{L}_1 + \ldots + \mathcal{L}_C)^*$ and $C$ is the number of word piece tokens.

## 4.3 CONVOLUTIONAL WFSTS

Thus far we have only discussed applications of WFSTs to combine the output of other models. Figure 6 demonstrates a convolutional WFST layer which can function as an arbitrary layer in a tensor-based architecture. Like a standard convolution, the WFST convolution is specified by a set of kernels, a receptive field size $k$, and a stride $s$. Each kernel, $\mathcal{K}_i$, is a WFST. Let the hidden units in a receptive field at position $t$ be given by $H_{t:t+k} \in \mathbb{R}^{d \times k}$. The receptive field graph, $\mathcal{R}_{H_{t:t+k}}$, is a linear graph with $k$ nodes and $d$ edges between consecutive nodes. The activations in $H_{t:t+k}$ yield the edge weights in the receptive field graph. The output of the $i$-th kernel applied to position $t$ is

$$H'_{i,t} = \underset{\boldsymbol{p} \in \mathcal{K}_i \circ \mathcal{R}_{H_{t:t+k}}}{\text{logadd}} s(\boldsymbol{p}), \tag{4}$$

Table 1: A comparison of dense transition graphs from $n = 0$ (no transitions) up to $n = 2$ (bi-grams). We report CER on the IAM validation set using letter tokens.

| $n$-gram | No blank | Forced blank | Optional blank |
|---|---|---|---|
| 0 | 18.2 | 8.9 | 6.4 |
| 1 | 17.7 | 8.5 | 6.3 |
| 2 | 9.4 | 6.7 | 6.4 |

Table 2: A comparison of bi-gram transition graphs with back-off and varying levels of pruning for letters and $1,000$ word pieces. We report CER on the IAM validation set and epoch time in seconds.

| Pruning | Letters | | Word Pieces | |
|---|---|---|---|---|
| | CER | Time (s) | CER | Time (s) |
| None | 6.4 | 544 | N/A | 17,939 |
| 0 | 6.8 | 249 | 22.2 | 683 |
| 10 | 6.5 | 202 | 19.8 | 204 |

the forward score of the receptive field graph composed with the kernel graph.

The structure and labels of the kernel WFSTs are problem specific. For example, the kernel transducers can be structured to impose a desired correspondence. We construct kernels which transduce lower-level tokens such as letters to higher level tokens like word pieces. Without any other constraints, the interpretation of the input as letter scores and the output as word piece scores is implicit in the structure of the graphs, and can be learned by the model up to any indistinguishable permutation. The kernel graphs themselves have edge weights which, since equation 4 is differentiable, can be initialized randomly and learned along with the other parameters of the model.

## 5 EXPERIMENTS

We study several of the algorithms from section 4 with experiments in offline handwriting recognition (HWR) and automatic speech recognition (ASR). For both applications, we use a variation of the time-depth separable (TDS) convolutional architecture (Hannun et al., 2019). A more detailed description of the network architectures, optimization procedures and data processing is in Appendix B. We report character error rate (CER) without the use of an external language model or lexical constraint. Code to reproduce our experiments is open-source and available at https://www.anonymized.com. All of the tensor based operations are on the GPU while the WFST operations are on the CPU. The WFST operations are executed in parallel over the examples in the batch.

**Offline Handwriting Recognition** To facilitate comparisons to prior academic studies, we test our approach on handwritten line recognition using the IAM database (Marti & Bunke, 2002). We use the standard training set and report results on the first validation set available with the data set distribution. The splits have changed from those used by prior work (Graves et al., 2008; Pham et al., 2014; Voigtlaender et al., 2016) which makes comparisons to such work less meaningful.

**Automatic Speech Recognition** Speech recognition experiments are performed on the LibriSpeech (Panayotov et al., 2015) and Wall Street Journal (WSJ) (Paul & Baker, 1992) corpora. For LibriSpeech, we train with the "train-clean-100" subset and report results on the clean validation set. For WSJ we train on the full corpus and report results on the "dev-93" validation set.

### 5.1 TRANSITIONS

Table 1 shows that including transitions can improve CER substantially, particularly when no blank token is used. We speculate that bigram transitions help in this case primarily because of the duration model they provide. As previously observed (Graves et al., 2006), with a blank token, the emissions of non-blank outputs spike at a single frame, and no duration model is needed. In Table 2, we see that pruning with back-off results in no discernible loss in accuracy over a dense transition model while yielding substantial run-time improvements. With 1000 word pieces, the epoch time using a graph with pruned bigrams is nearly two orders of magnitude faster than that of the dense graph.

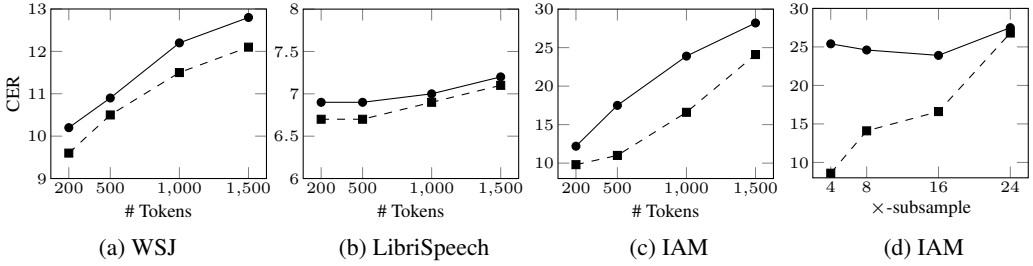

|  |  |  |  |
|---|---|---|---|
| (a) WSJ | (b) LibriSpeech | (c) IAM | (d) IAM |

Figure 7: Validation CER with (dashed) and without (solid) marginalization as a function of (a-c) the number of word pieces and (d) the overall sub-sample factor using 1000 word pieces.

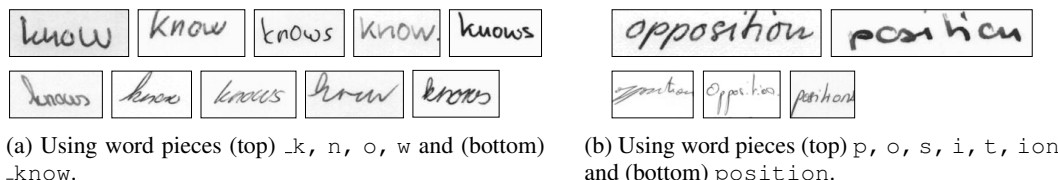

(a) Using word pieces (top) _k, n, o, w and (bottom) _know.

(b) Using word pieces (top) p, o, s, i, t, ion and (bottom) position.

Figure 8: The Viterbi word piece decomposition with 1000 tokens for (a) "_know" and (b) "position". The decompositions are computed on the training set images prior to cropping the words.

## 5.2 Marginalized Word Piece Decompositions

We experiment with marginalization over word piece decompositions in both ASR and HWR. We use the SentencePiece toolkit (Kudo & Richardson, 2018) to compute the token set from the training set text. For all datasets, larger token set sizes result in worse CER, though the effect is mild on LibriSpeech and more pronounced on IAM. Appendix figure 11 suggests this is primarily due to the different dataset sizes. Figures 7a, 7b, and 7c demonstrate that word piece marginalization recovers some of the accuracy lost when using larger token sets. We examine the Viterbi decomposition on the LibriSpeech training set. Appendix table 4 shows that in some cases the most frequently selected decomposition for a word is more phonologically plausible than the SentencePiece decomposition.

For IAM, we find that the model favors decompositions using lower-level output tokens but does still rely on higher-level word pieces in many cases. We also explore the interaction between sub-sampling in the network and marginalization. One benefit of marginalization is the ability to dynamically choose the decomposition suitable to a given frame rate as demonstrated by 7d. We examine the Viterbi decomposition for a few examples from the training set in figure 8. We see that a single model dynamically switches between different decompositions depending on the input. If the handwriting is tightly spaced, for example, a higher-level token is preferred.

## 5.3 Convolutional WFSTs

We experiment with the convolutional WFST layer on the IAM data set. The kernels in the WFST layer transduce letters to 200 word pieces. We use a CTC-style graph structure for the kernels and hence the input to the WFST has an additional dimension for the  token. Figure 9 gives and example of a kernel and receptive field WFST. We place the WFST convolution in between the third and fourth TDS groups. The model is trained to predict 1000 word pieces at the output layer. We compare the WFST convolution with a traditional convolution in the same position. Table 3 demonstrates that the WFST convolution outperforms the traditional convolution. The WFST convolution is also less expensive as the number of parameters and the number of operations needed scale with the lengths of the word pieces in letters rather than the number of input channels.

## 6 Conclusion

We have shown that combining automatic differentiation with WFSTs is not only possible, but a promising route to new learning algorithms and architectures. The design space of structured loss

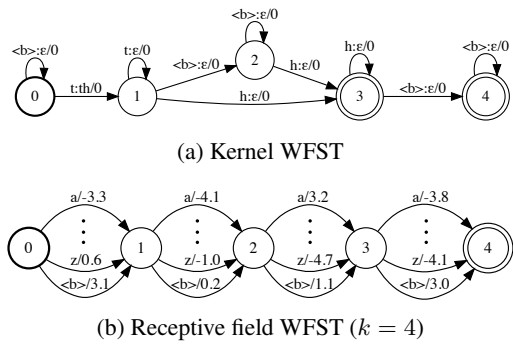

(a) Kernel WFST

(b) Receptive field WFST ($k = 4$)

Figure 9: Graphs for the convolutional WFST.

Table 3: A comparison of the convolutional WFST layer to a traditional convolution. We report the CER on the IAM validation set and compare the two layers in number of parameters and number of operations. Both convolutional layers have a kernel width $k = 5$, a stride of 4, $c_i = 80$ input channels, and $c_o = 200$ output channels.

| Convolution | CER | Params | Ops |
|---|---|---|---|
| WFST | 19.5 | 2,048 | $O(kwc_o)$ |
| Traditional | 20.7 | 79,000 | $O(kc_i c_o)$ |

functions that may be encoded and efficiently implemented through the use of differentiable WFSTs is vast. We have only begun to explore it.

We see many other related and exciting paths for future work in automatic differentiation with WFSTs. First, we aim to continue to bridge the gap between static computations with WFSTs while performing inference and dynamic computations with WFSTs during model training. This will require further optimizations and potentially approximations when integrating higher level lexical and language modeling constraints. Second, the use of WFSTs as a substitute for tensor-based layers in a deep architecture is intriguing. This may be more effective than traditional layers in imposing prior knowledge on the relationship between higher-level and lower-level representations or in discovering useful representations from data.

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

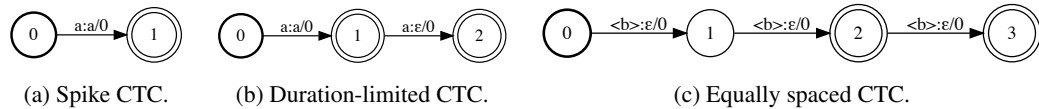

(a) Spike CTC.          (b) Duration-limited CTC.          (c) Equally spaced CTC.

Figure 10: A few individual token graphs used to construct variants of, for example, CTC. The individual token graphs are combined to create the overall token graph $\mathcal{T} = (\mathcal{T}_1 + \ldots + \mathcal{T}_C)^*$.

Hang Su, Gang Li, Dong Yu, and Frank Seide. Error back propagation for sequence training of context-dependent deep networks for conversational speech transcription. In *2013 IEEE International Conference on Acoustics, Speech and Signal Processing*, pp. 6664–6668. IEEE, 2013.

Seiya Tokui, Kenta Oono, and Shohei Hido. Chainer: a next-generation open source framework for deep learning. 2015.

Karel Veselỳ, Arnab Ghoshal, Lukás Burget, and Daniel Povey. Sequence-discriminative training of deep neural networks. 2013.

Paul Voigtlaender, Patrick Doetsch, and Hermann Ney. Handwriting recognition with large multidimensional long short-term memory recurrent neural networks. In *2016 15th International Conference on Frontiers in Handwriting Recognition (ICFHR)*, pp. 228–233. IEEE, 2016.

## A  ALGORITHMS

### A.1  REPEAT TOKENS IN CTC

The blank token in CTC () also serves to disambiguate consecutive repeat tokens in the output from a single token corresponding to two input frames. Enforcing this construction in the WFST framework requires keeping track of the previously generated token and only allowing transitions on a new token or on . Let $\mathcal{T}_1, \ldots, \mathcal{T}_C$ be the individual token graphs, including the blank token graph as in figure 3a. The full token graph to disallow repeat consecutive transitions is given by

$$\mathcal{T} = \left( \sum_{i=1}^{C} \sum_{j=1, j \neq i}^{C} \mathcal{T}_i \mathcal{T}_j \right)^*. \tag{5}$$

In practice, constructing the graph with equation 5 will result in more states and arcs than are needed. Instead, we construct the graph directly. However, we note that these operations may be used along with $\epsilon$ removal and state minimization to keep the resulting token graph $\mathcal{T}$ small.

### A.2  PRIORS WITH CTC

The individual token graphs in CTC do not assume much about the duration or spacing of the tokens at the input level. The graph in figure 2a states only that non-blank tokens must align to at least one frame. The graph in figure 3a states that the  token is optional.

The differentiable WFST framework simplifies the construction of variations of these token-level graphs with potentially useful alternative assumptions. For example, we can construct a "spike" CTC which only allows a single repetition of a non-blank label by using the graph in figure 10a. Similarly, a "duration-limited" CTC could use a token graph as in figure 10b. In this case, the duration of a token is limited to be between one and two input frames. We could specify a range of allowed distances between non-blank tokens using the blank token graph in figure 10c. These may all be mixed and matched on a per-token basis.

## B  EXPERIMENTAL SETUP

In both handwriting recognition and speech recognition experiments, we use a variation of the time-depth separable (TDS) convolutional architecture (Hannun et al., 2019). The code and settings needed to reproduce our experiments are available at https://www.anonymized.com.

Table 4: A comparison of the most frequent decomposition to the SentencePiece decomposition for a given word. The words were selected by first computing the Viterbi decomposition for each example in the LibriSpeech training set. We then chose words for which the most frequently selected decomposition was different from the SentencePiece decomposition. The counts are the occurrences of each decomposition in the Viterbi paths on the training set. In some cases the most frequent decomposition appears to be more phonologically plausible than the SentencePiece decomposition.

| Word | Most common | | SentencePiece | |
|------|-------------|-------|---------------|-------|
| | Decomposition | Count | Decomposition | Count |
| able | `_a, ble` | 121 | `_, able` | 88 |
| wind | `_w, in, d` | 90 | `_wi, n, d` | 72 |
| single | `_si, ng, le` | 65 | `_, s, ing, le` | 40 |
| move | `_mo, ve` | 54 | `_mov, e` | 47 |
| ring | `_ri, ng` | 48 | `_r, ing` | 23 |

## B.1 HANDWRITING RECOGNITION

For experiments in HWR, we propose a variant of the TDS model which preserves the 2D structure in the image. The 2D TDS block first applies a 3D convolution with kernel size $1 \times k_h \times k_w$ on a 4-dimensional input of size $c \times d \times h \times w$ where $c$ and $d$ are the separated channel and depth dimensions and $w$ and $h$ are the height and width of the image respectively. Following the 3D convolution, is a $1 \times 1$ convolution with $c \times d$ input and output channels. We apply dropout, the ReLU non-linearity and residual connections as in the original TDS architecture. In place of layer normalization, we use 2D instance normalization (Isola et al., 2017) with a learned affine transformation.

Unless otherwise noted, we use 4 groups of three TDS blocks each. In between each TDS block we apply a standard 2D convolution and optionally sub-sample the image in the height and width dimensions. For letter-based experiments we sub-sample the height by 2 and the width by 2 only after the first two groups. For word piece experiments we sub-sample both the height and width by a factor of 2 after every group for an overall sub-sampling factor of $16\times$ in each dimension. We use a uniform kernel size of $5 \times 7$. We start the number of channels ($c \times d$) at 8 and increase it to 16, 32 and 64 in between each TDS group.

For optimization we use a simple stochastic gradient descent with an initial learning rate of $10^{-1}$ which is reduced every 100 epochs by a factor of 2. All models are trained for 400 epochs and we save the model which achieves the best results on the validation set during training. We use a mini-batch size of 32 with distributed data parallel training.

Bounding boxes for the line-level segmentation for each page are provided along with the ground truth texts. We perform no other data pre-processing other than a simple normalization, removing the global pixel mean and standard deviation from each image. We use three forms of data augmentation–a random resize and crop, a random rotation, and a random color change.

## B.2 SPEECH RECOGNITION

For experiments in speech recognition, we use a 1D TDS architecture with 3 groups each containing 5 TDS blocks. In between each group we sub-sample the input in the time-dimension by a factor of 2 for an overall sub-sampling factor of 8. The TDS blocks all use a kernel width of 5. The first group has 4 channels, and we double the number of channels in between each group.

For optimization we use stochastic gradient descent. Initial learning rates are tuned for each model type in the range $10^{-2}$ to 1. The learning rate is reduced every 100 epochs by a factor of 2. All models are trained for 400 epochs and we save the model which achieves the best results on the validation set during training. We use a mini-batch size of 16 with distributed data parallel training.

For features, we use 80 log-mel-scale filter banks on windows of 20 milliseconds every 10 milliseconds. We normalize individual examples by removing the mean and dividing by the standard deviation. We apply SpecAugment for regularization following the "LD" policy but without time-warping (Park et al., 2019).

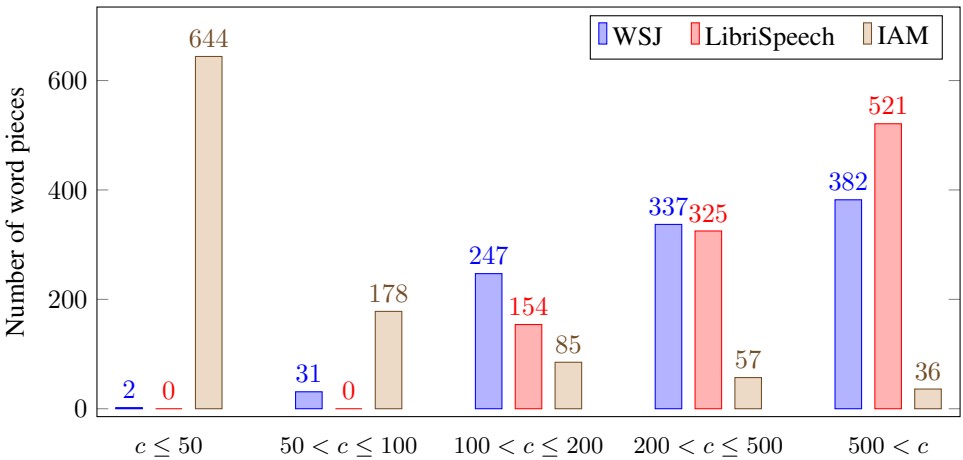

Figure 11: The number of word pieces with occurrences $c$ in the given range. For each dataset we use 1,000 word piece tokens and count the number of occurrences in the training text using the decomposition for each word provided by SentencePiece. The WSJ training text contains 639k words, the LibriSpeech training text has 990K words and the IAM training text has only 54K words.

