# OpenReview forum: "Differentiable Weighted Finite-State Transducers"
_ICLR.cc/2021/Conference — Reject_

### Official Review · AnonReviewer4 · 2020-10-20
**Good & interesting paper, though missing important details; it promises more than it delivers.**

**Rating:** 6
**Confidence:** 5

**Review:**

### Overall comments

This is a well-written paper describing the incorporation of WFSTs into an "auto-diff" framework for Deep Learning. The central point is that by providing partial derivatives, in addition to the standard forward scores, to the major WFST operations, one can embed those WFST operations into a gradient-based deep learning framework as just another differentiable module. Given the continued demonstrated utility of WFSTs in the speech and natural language community, many researchers will welcome the motivation. The paper details how some specific WFST models can help implement models such as CTC or ASG, presents a less-clear application to convolutional models, and presents results evaluating the use of differentiable WFSTs for tasks in ASR and handwriting recognition.

The paper is a good read, but the central contribution isn't really detailed. A good overview of WFSTs and their utility is given, but that is somewhat orthogonal to the central contribution of "we made an auto-diff version of these tools". Section 3 gives a more or less standard outline of the WFST framework, but only in Section 3.2 is auto-differentiation discussed, and it's a very bare-bones description, along the lines of, "we did it." The use of a gradient graph representing gradients/jacobians as WFST weights makes sense, but the specifics of implementing this for different WFST operations aren't really discussed. The reader doesn't have a sense of the implementation issues, or of any compute/memory issues involved. There is no sense of how efficiently the operations will run on GPUs. Since e.g. OpenFST operations such as ShortestDistance() and Divide() can be used to compute arc posteriors for ASR lattices in the context of MMI or sMBR for ASR sequence training, and since partial derivatives of the MMI/sMBR sequence level loss functions correspond to accumulated lattice arc posteriors, the reader who is knowledgeable about work in that area may wonder if the work detailed here also used WFST operations to define the derivatives themselves, or if a more direct/efficient implementation was used.

Though the application to CTC and ASG is very clear, the convolutional WFST is much less clear to me. If a kernel is a WFST, what is the sequence model? I.e. in Eq (4) , what are the sequences p? I think sections 4.3 and 5.3 should expand on the convolutional models used, and detail both the standard convolutional model and the WFST model used. I think the paper assumes the reader is familiar with the TDS / convnet seq2seq models previously proposed, but I for one am not.

More specific comments:

### References/background:

This well-known work by Jason Eisner is in my mind closely related to the ideas in this paper:

https://www.aclweb.org/anthology/P02-1001.pdf.

"For example, Kaldi, a commonly used
toolkit for automatic speech recognition, uses WFSTs extensively, but in most cases for inference or
to estimate the parameters of shallow models (Povey et al., 2011). In some cases, WFSTs are used
statically to incorporate fixed lattices in discriminative sequence criteria (Vesely et al., 2013). "

All state-of-the-art hybrid ASR systems use some form of sequence training based on ASR lattices and criteria such as MMI or sMBR. Just a few references (in addition to the Vesely reference):

https://ieeexplore.ieee.org/stamp/stamp.jsp?arnumber=4960445
https://www.isca-speech.org/archive/archive_papers/interspeech_2012/i12_0010.pdf
https://ieeexplore.ieee.org/document/6638951
https://www.isca-speech.org/archive/archive_papers/interspeech_2014/i14_1224.pdf

"Implementations of sequence criteria in end-to-end style training are typically hand-crafted with
careful consideration for speed (Amodei et al., 2016; Collobert et al., 2019; Povey et al., 2016)."

Though that is true for new models such as ASG and Lattice-Free MMI, for LAS and RNN-T, the models are locally normalized and fully conditioned on past history, so the baseline can be thought of as already sequence-trained (there is no need for a normalizing global score), which simplifies things. But by the same token, those models being fully conditioned on the past symbol history don't in fact allow the use of lattices (without limited history approximations) for criteria such as WMBR, so typical approaches use N-best lists. I'll leave it up to the authors how, or if, they want to acknowledge this point in their draft, it's really just a comment -- just to say that for some in the "end-to-end" world, WFSTs are less relevant than to others, depending on the nature of their models. Some references along those lines:

https://ieeexplore.ieee.org/stamp/stamp.jsp?arnumber=8461809 (MBR of LAS models)
https://arxiv.org/pdf/1911.12487.pdf (MBR of RNN-T models)

### Other comments

"Graphs": this is never actually defined afaict. Is it synonymous to "WFST"?

"Forward and Viterbi Score The forward score of T is logadd(p,r)∈A s(p, r). Similarly, the Viterbi
score of T is max(p,r)∈T s(p, r)."

Why is the Forward score defined over an Acceptor, while the Viterbi score is defined over a Transducer?

Section (4) Eq. (1): just an FYI, a lot of readers in the ASR community would recognize A and Z as "numerator" and "denominator" for lattice-based sequence training, see e.g. https://www.isca-speech.org/archive/archive_papers/icslp_1996/i96_0018.pdf.

"The primary difference between ASG and CTC is the inclusion of the blank token graph": isn't the primary difference (in addition to the one cited), the use in ASG of a normalizing score, while in CTC the outputs are all locally normalized? (There is no normalizing score in CTC , at least not in the same sense as there is in ASG).

Section 5.3, "TDS" is mentioned with no reference and no explanation of what it is an abbreviation for, and no summary of what it is.

"The kernel transducers can be structured to impose a desired correspondence. For example, we
construct kernels which map lower-level tokens such as letters to higher level tokens like word
pieces. Without any other constraints, the interpretation of the input as letter scores and the output
as word piece scores is implicit in the structure of the graphs, and can be learned by the model up
to any indistinguishable permutation"

This relates to my points at the start -- this description is much too compact, I really have to strain to imagine the specifics of what the authors are talking about here. Also, what is the standard (non-convolutional) model here?

---

> ### Author Response · Authors · 2020-11-20
> **Responses to AnonReviewer4**
>
> Thank you for your comments. Our detailed responses are below the original question or comment in italics.
>
> *Q1: The paper is a good read, but the central contribution isn't really detailed.*
>
> Please see the top-level comment under “The goal of this work” as well as the list of contributions in the introduction of the paper. Our paper focuses on the (novel) algorithms one can easily develop using a WFST with automatic differentiation to highlight the power of this concept. We deliberately chose not to focus on the implementation details, which while we agree are very important we felt 1) are still subject to change depending on how the framework is used 2) would require a separate technical report to give reasonable detail.
>
> *Q2: Though the application to CTC and ASG is very clear, the convolutional WFST is much less clear to me. If a kernel is a WFST, what is the sequence model? I.e. in Eq (4) , what are the sequences p?*
>
> We have added more details on this in the manuscript along with two figures illustrating the graphs in section 5.3. The sequence model still uses the same CTC loss at the output as all of the other models. The convolutional WFST layer should just be viewed as a replacement for a standard convolution in the network itself. The sequences $p$ in equation (4) are any valid paths in the composition of the kernel and receptive field graph. In our application this would be valid mappings of letters to word pieces. For example the path `ttthh<b><b>e<b>`  is a valid path mapping from letters to the word piece `"the" when the receptive field has 9 frames.
>
> *Q3: I think sections 4.3 and 5.3 should expand on the convolutional models used, and detail both the standard convolutional model and the WFST model used. I think the paper assumes the reader is familiar with the TDS / convnet seq2seq models previously proposed, but I for one am not.*
>
> We have added more details on the convolutional WFST. We also introduce the TDS model in section 5 and point to appendix B which contains a much more complete description.
>
> *Q4: Additional references.*
>
> Thank you for pointing out the extra references. We have added these to the manuscript.
>
> *Q5: "Graphs": this is never actually defined afaict. Is it synonymous to "WFST"?*
>
> Yes, we use the term "Graph" and "WFST" interchangeably. We have defined "Graph" more clearly in the manuscript.
>
> *Q6: Why is the Forward score defined over an Acceptor, while the Viterbi score is defined over a Transducer?*
>
> This is a typo, thanks for catching the mistake.
>
> *Q7: "The primary difference between ASG and CTC is the inclusion of the blank token graph": isn't the primary difference (in addition to the one cited), the use in ASG of a normalizing score, while in CTC the outputs are all locally normalized?*
>
> Thank you for pointing this out, we agree the blank token is not the only primary difference, we have updated this in the manuscript. We view the two primary differences between CTC and ASG as 1) the inclusion of the blank token and 2) the use of bigram transitions. CTC is locally normalized in as much as the global normalization term factorizes easily because it does not have transitions. However, it is also correct to say that CTC is globally normalized (See section 4.3 of [1]). We view the local normalization more as a result of the fact conditional independence assumptions of CTC rather than as a difference in and of itself.
>
> *Q8: Section 5.3, "TDS" is mentioned with no reference and no explanation of what it is an abbreviation for, and no summary of what it is.*
>
> Thank you for catching this, we added an introduction to the abbreviation at the top of section 5. A much more complete description of the TDS model is also available in the appendix B.
>
> *Q9: This relates to my points at the start -- this description is much too compact, I really have to strain to imagine the specifics of what the authors are talking about here. Also, what is the standard (non-convolutional) model here?*
>
> We have added details to the section on the WFST convolution in an attempt to make it clearer.
>
> [1] The Label Bias Problem, Awni Hannun, https://awnihannun.com/writing/label_bias.pdf

---

### Official Review · AnonReviewer2 · 2020-10-27
**Interesting software with limited scientific novelty**

**Rating:** 4
**Confidence:** 5

**Review:**

This is a paper describing a software package for common calculations on WFSTs (like forward score or best path score), in a way that they are differentiable w.r.t. the arc weights.

It also contains several examples of WFSTs and WFSAs specifically for speech recognition or handwriting recognition, like for the CTC or ASG topology.

The scientific novelty in the paper is very limited. The scientific new contribution is a convolutional WFST layer.

Random notes and questions:

- The K2 project (https://github.com/k2-fsa/k2) is a very popular attempt in this space which seems to have very similar goals. Specifically it has efficient implementations to calculate the best path score or forward score and is differentiable w.r.t. the arc weights. What are the differences?
- The PyChain project (https://arxiv.org/abs/2005.09824, https://github.com/YiwenShaoStephen/pychain) also seems very related. The FSA structure is created by Kaldi, exported as OpenFST, and then imported as via OpenFST bindings in PyChain. PyChain then has an efficient GPU-based implementation of the forward-backward algorithm which gets any such FSA. The arc weights are dynamically filled in via posteriors from a NN, and the fwd-bwd score can then be used as the gradient. This is slightly less generic, as the arc weights are assumed to be coming from posteriors. But despite that, what are the differences?
- The RETURNN project (https://github.com/rwth-i6/returnn) also has a similar generic implementation, which gets in any generic FSA, and then calculates the Viterbi path/score or forward-backward (Baum Welch algorithm), and the arc weights are given by NN posteriors. This was presented in the paper ["CTC in the Context of Generalized Full-Sum HMM Training"](https://www-i6.informatik.rwth-aachen.de/publications/download/1035/ZeyerAlbertBeckEugenSchl%FCterRalfNeyHermann--CTCintheContextofGeneralizedFull-SumHMMTraining--2017.pdf). This is a fast batched GPU implementation.
- It is stated that Kaldi mostly uses WFSTs for inference. This is wrong. Esp in the case of LF-MMI training, it uses WFSTs in training.
- Details about how the fwd-bwd computation (or other computations) are done are missing. E.g. is this implemented in CUDA and runs on GPU? How exactly? At least some details about this should be given.
- The WFST arcs (from,to edges), arc weights, list of final states is all in GPU memory? Does it (including the algos like fwd-bwd) support batching, i.e. having multiple WFSTs in one batch? This is specifically important for efficient training.
- How fast is it? E.g compared to the existing implementation above, or other existing implementations for CTC.
- Is the WFST always represented statically, or does it support dynamic composition, which would be done on-the-fly when traversing the graph (like WFST decoders in speech recognition usually do it)? If it is always static, it might become very big, e.g. when combined with a 5gram LM or so, and then might not fit into memory. How is this solved?
- In e.g. TensorFlow, you can easily explicitly calculate the forward scores in a while loop by explicitly iterating through allowed transitions. You don't need to formulate it as a WFST. This would also be fully differentiable, also supports full GPU-based calculation, and is very straight-forward. What is the advantage of the WFST formulation? The explicit calculation can be even more flexible, like only calculate arc weights which are really needed (e.g. in beam search).
- The motivation for the convolutional transducer is not exactly clear.
- The exact calculation/definition of the convolutional transducer is not clear. What exactly is the structure of the WFST, and how are the arc weights defined?
- It is stated that IAM CER can not be compared to the literature because a different validation set is used. Why? Why not use the right validation set such that it can be compared? The CER also looks quite high, but maybe the validation set is much harder? Or the model is very weak compared to current state-of-the-art?
- The IAM CER in Table 3 is esp very high, much higher than in Table 1 or 2. Why is that?
- Speech recognition experiments are performed, but no word-error-rate (WER) is reported at all, which is the standard measure for speech recognition. Why? It looks like it deliberately is not supposed to be compared to other results from the literature?


Pros:

- The software package and capabilities sound very promising.

Cons:

- It misses several very related works (as outlined above).
- Some important details about the implementation are missing (as outlined above).
- The motivation (and exact definition) of the convolutional WFST are unclear.
- The experimental section is very short, lacks in depth, and misses comparisons to the literature.

Summary:

Overall, the software looks interesting. However, the scientific novelty is limited, and also the experimental section has several problems. Maybe ICLR is not the right conference for this paper? Or otherwise the scientific contributions and experimental section should be improved. E.g. more tasks and experiments should be done where it is useful to have such a generic differentiable WFST software, which is currently hard to implement otherwise. And the experiments should have fair comparisons to the literature. In general, there are also several problems in the related work section.

---

> ### Author Response · Authors · 2020-11-20
> **Responses to AnonReviewer2**
>
> Thank you for your comments. Our detailed responses are below the original question or comment in italics.
>
> *Q1: Differences between K2, PyChain and RETURNN.*
>
> Please see the discussion in the top-level comment under “Novelty”. At a high-level, K2 is very similar to our framework and developed (though not yet released) along a similar timeline. PyChain is very specific to LF-MMI. Similarly, while RETURNN may use WFSTs under the hood in certain cases, it does not in any general sense allow for automatic differentiation with WFSTs and their operations.
>
> *Q2: It is stated that Kaldi mostly uses WFSTs for inference. This is wrong. Esp in the case of LF-MMI training, it uses WFSTs in training.*
>
> Kaldi uses WFSTs for LF-MMI training, but not in a generic way with automatic differentiation. This is one motivation for the development of K2 (and our work) as the LF-MMI implementation in Kaldi is quite specialized.
>
> *Q3: Details about how the fwd-bwd computation (or other computations) are done are missing. E.g. is this implemented in CUDA and runs on GPU? How exactly? At least some details about this should be given.*
>
> Thank you for the comment. We have provided more details on this in the manuscript. All of the WFST operations (forward, viterbi, compose, epsilon removal, etc.) are currently on the CPU.
>
> *Q4: The WFST arcs (from,to edges), arc weights, list of final states is all in GPU memory? Does it (including the algos like fwd-bwd) support batching, i.e. having multiple WFSTs in one batch? This is specifically important for efficient training.*
>
> We have provided more details on this in the manuscript. Our implementation of WFST operations does support batching and the WFST operations are implemented in parallel over examples in the batch (which takes place on the CPU).
>
> *Q5: Is the WFST always represented statically, or does it support dynamic composition?*
>
> We do not yet support lazy composition. If this turns out to be an important feature needed during training we will definitely prioritize implementing it.
>
> *Q6: In e.g. TensorFlow, you can easily explicitly calculate the forward scores in a while loop by explicitly iterating through allowed transitions. You don't need to formulate it as a WFST. This would also be fully differentiable, also supports full GPU-based calculation, and is very straight-forward. What is the advantage of the WFST formulation?*
>
> The WFST formulation support many more operations than just “forward score” including, most importantly, composition. Our framework provides a calculus on WFSTs which allows for the construction of complex WFSTs from simpler ones in a fully-differentiable way. Furthermore, even forward on a generic non-linear WFST is not as straightforward to do in TensorFlow as the reviewer suggests.
>
> *Q7: The motivation for the convolutional transducer is not exactly clear. The exact calculation/definition of the convolutional transducer is not clear. What exactly is the structure of the WFST, and how are the arc weights defined?*
>
> Thank you for the comment, we have added more details on the convolutional WFST in the manuscript.
>
> *Q8: It is stated that IAM CER can not be compared to the literature because a different validation set is used. Why? Why not use the right validation set such that it can be compared? The CER also looks quite high, but maybe the validation set is much harder? Or the model is very weak compared to current state-of-the-art?*
>
> We were unfortunately unable to find the original train-validation-test split on the [IAM website](https://fki.tic.heia-fr.ch/databases/iam-handwriting-database) or through references in prior work. The CER for our character-based model is comparable to the best reported in the literature [1]. We report 6.3 CER on validation and Kang, et al. [1] report 7.6 CER on test.
>
> *Q9: The IAM CER in Table 3 is esp very high, much higher than in Table 1 or 2. Why is that?*
>
> Our word piece models for this task are in general quite a bit worse than the character based models and we give some explanation for this in section 5.2 and in the appendix figure 10.
>
> *Q10: Speech recognition experiments are performed, but no word-error-rate (WER) is reported at all, which is the standard measure for speech recognition. Why? It looks like it deliberately is not supposed to be compared to other results from the literature?*
>
> Please see the second bullet point in the top-level comment under “The goal of this work”. This is not a deliberate attempt to thwart comparisons to prior work. We did not implement this given the primary goal of this work as discussed in the top-level comment.
>
> [1] Pay Attention to What You Read: Non-recurrent Handwritten Text-Line Recognition, Lei Kang, Pau Riba, Marçal Rusiñol, Alicia Fornés, Mauricio Villegas, https://arxiv.org/abs/2005.13044

---

### Official Review · AnonReviewer3 · 2020-10-28
**An interesting graph transformer networks revival**

**Rating:** 5
**Confidence:** 4

**Review:**

This paper presents how weighted finite-state transducers (WFST) and a few common operations performed on them can be integrated in a differentiable model, and therefore contribute to the training of complete systems. The authors propose a few case studies, mainly in language applications, where the WFSTs are used to compute a sequence-level loss function, to keep ambiguity and let the model decide word-peices decomposition, or gracefully replace convolution layers. The code associated with the presented methods will be available.

The benefits brought by the availability of WFST-based differentiable operation in deep learning libraries are clear and very relevant. As the authors mention, it would unlock a wealth of possibilities in the design of loss functions and other operations in language-related applications (but not only) and allow to more easily create end-to-end systems.

However, the idea in itself is not entirely novel. As mentioned in the related work section, this idea has been extensively explored in the past, and a good example of this are the graph transformer networks (GTNs). The corresponding paper(s) cover the same kind of operations described here if not more, and it is not clear how this paper brings more on that topic, other than putting them in the context of loss functions used today. The differentiability of the selected operations in the selected semiring is pretty straight-forward and already implemented in some frameworks for special cases. As rightfully mentioned by the authors, Kaldi implements sequence-discriminative loss functions, either lattice-based or lattice-free, based on WFSTs and on GPU. PyChain implements it too, and I believe Google mentioned at several occasions that their implementation of the CTC loss was based on WFSTs.

The novelty therefore does not lie so much in the differentiability of WFSTs nor in their integration into the training procedure as in the implementation of an efficient way to integrate them in a generic fashion for GPU training. The paper should focus more on that aspect, which would indeed be very interesting for the community, reviving the very interesting and not-so-much exploited GTN idea.

Regarding the applications of the method to speech and handwriting recognition and word-piece selection, they are quite relevant for the scope of this paper. Although the level of details on CTC and ASG for example would be sufficient for application-specific conferences, I feel like a reminder of how and why these loss functions are computed would be nice for ICLR since not all readers may be familiar with these losses.

Regarding the implementation, which to me is the main contribution of this paper, I would expect more details about runtime and efficiency compared to other implementation of CTC and ASG for example, which by the way would not be too hard to modify to include the proposed variations (even though it would be reimplementing them for each new case, when the proposed method would be generic, which is indeed a very nice thing to have!). It would also be interesting to see more the impact of the graph size and structure, understand more how the epsilon transitions are handled and the derivative computed.

Regarding the experiments, a comparison with the state-of-the-art would be interesting. I understand that the page limit is tight and does not allow to present the models in details, but the reference to the paper presenting TDS in the main text at least might help the reader understand better section 5.3, which is difficult to follow when one is not familiar with that architecture.

Overall, the paper is well-written, easy to read when one is familiar with the presented loss functions. The idea is attractive and the implementation would be very beneficial for the community. However, the novelty of the idea itself is very limited and the improvement over the GTN idea is not clearly stated. The contribution is in my opinion more related to an efficient implementation, but that part is not really described here and lacks an empirical study of its performance. Although I'd be happy to use the implementation provided by the authors and understand the huge benefit of its availability, I do not see this paper above acceptance threshold for ICLR.

---

> ### Author Response · Authors · 2020-11-20
> **Responses to AnonReviewer3**
>
> Thank you for your comments. Our detailed responses are below the original question or comment in italics.
>
> *Q1: However, the idea in itself is not entirely novel. As mentioned in the related work section, this idea has been extensively explored in the past, and a good example of this are the graph transformer networks (GTNs).*
>
> Please see the top-level response under "Novelty". While the prior GTN work is similar to the work here, some differences include:
>
> - We present several novel algorithms demonstrating the utility of a framework for differentiable WFSTs.
> - We provide reasonably efficient and open-source implementation which (to the best of our knowledge) prior to this work did not exist.
>
> *Q2: The differentiability of the selected operations in the selected semiring is pretty straight-forward and already implemented in some frameworks for special cases.*
>
> We agree some of these operations are implemented as special cases; however, we point out that the generality of our framework is important to leverage it to design new algorithms. We make very few assumptions about the WFST structure and provide many of the important operations including composition and epsilon removal.
>
> *Q4: As rightfully mentioned by the authors, Kaldi implements sequence-discriminative loss functions, either lattice-based or lattice-free, based on WFSTs and on GPU. PyChain implements it too, and I believe Google mentioned at several occasions that their implementation of the CTC loss was based on WFSTs. The novelty therefore does not lie so much in the differentiability of WFSTs nor in their integration into the training procedure...*
>
> The implementation of the examples the reviewer listed (Kaldi LF-MMI, PyChain, Google CTC) do not rely solely on WFSTs with generic WFST operations and automatic differentiation. These are all specialized implementations which may use WFSTs to do part of the computation but in general are highly specialized to the loss function implementation and hence difficult to change.
>
> *Q5: Although the level of details on CTC and ASG for example would be sufficient for application-specific conferences, I feel like a reminder of how and why these loss functions are computed would be nice for ICLR since not all readers may be familiar with these losses.*
>
> Thank you for the comment, we have added some more detail on these loss functions. Since we do not have enough space to add that much detail we also provided references for background material for those unfamiliar with these loss functions.
>
> *Q6: Regarding the implementation, which to me is the main contribution of this paper, I would expect more details about runtime and efficiency compared to other implementation of CTC and ASG for example... It would also be interesting to see more the impact of the graph size and structure, understand more how the epsilon transitions are handled and the derivative computed.*
>
> Please see the first bullet point in the top-level comment under “The goal of this work”. We agree with the reviewer that this is worth a detailed discussion, and we have every intention to do so in follow-on work as the implementation details solidify.
>
> *Q7: Regarding the experiments, a comparison with the state-of-the-art would be interesting.*
>
> Please see the second bullet point in the top-level comment under “The goal of this work”.

---

### Official Review · AnonReviewer1 · 2020-10-31
**A library for Differentiable Weighted Finite-State Transducers**

**Rating:** 6
**Confidence:** 5

**Review:**

Summary:

The authors introduce a  library for differential weighted finite-state transducers. WFST are commonly used in speech or handwriting recognition systems but are generally not trained jointly with the deep neural networks components such as ConvNN. This is not due to theoretical limitation of WFST but rather to a lack of available implementation and the need of important computational power to train them. The authors show that this new library can be used to encode the ASG criterion, by combining the emission graph (coming from a NN for example), the token graph (base recognition units) and the label graph (the sequence annotation) on one hand and the emission graph and a language model graph on the other hand. The authors show how word pieces decomposition can be learnt through marginalisation. Finally, convolution Wfst are rapidly presented. Preliminary experiments are reported on wSj data base for speech recognition and IAM database for handwriting recognition.


##########################################################################

Reasons for score:


 I am very pleased to see an implementation of the GTN approach which has been proposed more than 20 years ago. WFST approaches have been shown to be more effective (and more elegant) than ad-hoc implementation for both speech and handwriting recognition. If efficient, this library will certainly have a major impact on future ASR and HTR systems. However, implementation details are not given or explained and experiments are still preliminary. Despite its importance and impact, this work seems to be in a too early stage to be accepted to ICLR this year.



##########################################################################Pros:

Pros:

* first implementation of a differentiable WFST library
* experiments both on ASR and HTR with interesting results for learning WFST parameters
* a new convolutional WFST is introduced


##########################################################################

Cons:

* we dont' know to what extent the operations on WFST needed to build a real ASR/HTR application  are available (determinisation, minimization,  weight pushing, etc)
* ASR/HTR systems are not compared to state of the art, to measure the remaining progress to reach SOTA.
* As said by the authors, include WFST in a differentiable stack of layers needs a lot of computation. Is it trackable for large scale systems ? Table 2 gives epoch times for 1000 word pieces (which is small) and for bigrams only. Is it on TPU or CPU ?
* the section 4 on learning algorithms is not very generic as only an implementation of ASG is first presented then a comparison to CTC.
* section 4.3 on conv. WFST is too short to be really understand the proposed model. Maybe this part should be dropped to leave more room to basic algorithms presentation.

---

> ### Author Response · Authors · 2020-11-20
> **Responses to AnonReviewer1**
>
> Thank you for the comments.  Our detailed responses are below the original question or comment in italics.
>
> *Q1: Implementation details are not given or explained and experiments are still preliminary.*
>
> We have added more details on the convolutional WFST layer as this was a common point of feedback amongst the reviewers. We are also happy to provide any additional implementation details that the reviewer finds to be missing. We would also like to point out that the entire codebase will be made available to reproduce the experiments exactly as reported in the paper.
>
> *Q2: We don’t know to what extent the operations on WFST needed to build a real ASR/HTR application are available (determinization, minimization, weight pushing, etc).*
>
> We currently support composition with epsilon transitions, shortest path in the log and tropical semirings, epsilon removal and a few others. We do not yet have an implementation of determinization or minimization. These are on our roadmap but at the moment operations used during model training are taking larger precedence.
>
> *Q3: ASR/HTR systems are not compared to state of the art, to measure the remaining progress to reach SOTA.*
>
> Please see the second bullet discussed at the top.
>
> *Q4: As said by the authors, include WFST in a differentiable stack of layers needs a lot of computation. Is it trackable for large scale systems? Is it on TPU or CPU?*
>
> Yes this is indeed quite slow and may not be tractable for very large systems as is. Our implementation is on the CPU so it may be made much faster on a more parallel processor. We are investigating this. While we do example/batch-level parallelization, the layer itself can be parallelized at a lower-level which could make it much faster.
>
> *Q5: The section 4 on learning algorithms is not very generic as only an implementation of ASG is first presented then a comparison to CTC.*
>
> We agree with the reviewer that there are certainly loss functions which do not fit the paradigm we describe in this section (i.e. as the difference between the constrained and unconstrained graph). We did not intend for this to be a fully general discussion of algorithms one can implement with WFSTs.
>
> *Q6: Section 4.3 on conv. WFST is too short to be really understand the proposed model. Maybe this part should be dropped to leave more room to basic algorithms presentation.*
>
> Thank you, we have added details to this section in an attempt to make it clearer.

---

### Author Response · Authors · 2020-11-20
**Top level response to reviewers**

Dear reviewers,

Thank you for your constructive feedback. We also very much appreciate the positive feedback including comments such as this work may “*have a major impact on future ASR and HTR systems*”, could “*unlock a wealth of possibilities in the design of loss functions and other operations in language-related applications*”, and the “*idea is attractive and the implementation would be very beneficial for the community*”.

To start, we would like to address some comments that are common amongst the reviewers.

**The goal of this work:**

The intention of this research is to show that the idea of fully differentiable WFSTs in an easy to use framework allows for rapid experimentation of existing and novel (and hopefully interesting) algorithms. Any feedback on how to show this more clearly is much appreciated.

Some of the reviewers asked for more details on the implementation of the framework itself and state-of-the-art results. While we have attempted to provide some details here, we also would like to point out that these are not the primary goal of the work.

Specifically:

1. We do not intend to provide a detailed discussion of the design and implementation of the framework. We completely agree with suggestions that this is important, but we leave this as a separate goal which we intend to pursue in follow-on work. In part, we did not do this yet because we believe the underlying design may change substantially depending on how the framework is used.

2. We did not intend to give state-of-the-art results for either the handwriting or speech recognition benchmarks. We tuned our baselines to a point that we felt the results to be compelling. However, to achieve state-of-the-art would require substantially more tuning and implementation that is orthogonal to the primary goal of this research.

**Novelty:**

We see a trend in the comments that a framework for fully differentiable WFSTs is not novel. As far as we know, there are no other frameworks which implement the set of core algorithms (compose, forward, viterbi, epsilon removal, etc.) with automatic differentiation. The only exception to this is the not yet released Kaldi “K2” project. As far as we can tell our framework and K2 are on a very similar timeline, both starting development in the Winter/Spring of 2020.  We only learned about the K2 project around the time we were submitting this paper to ICLR.

We would also like to point out that the algorithms we present are novel. For example, we are not aware of prior work exploring CTC with transitions, marginalized word piece decompositions or convolutional WFSTs. However, the novelty of these algorithms are not the main goal. Rather, they are intended to demonstrate the core novelty of this work which (in a meta-sense) is the fact that one can use the framework to rapidly design new algorithms.

---

### Decision · Program_Chairs · 2021-01-07
**Final Decision**

**Decision:**

Reject

**Comment:**

This paper introduces a framework for automatic differentiation with weighted finite-state transducers (WFSTs), which would allow user-specified graphs in structured output prediction tasks and easy plug-and-play of graphs through the composition operation (demonstrated with variants of CTC). The authors demonstrated their framework on the OCR and ASR domains, which are important application scenarios. All reviewers agree the work is useful and can potentially be significant. However, the reviewers think the paper needs more discussions of similar/parallel work and the key differences from them, and clear description of the novelty in terms of either machine learning insights or algorithmic implementations.  We understand that this may be an implementation-heavy work, but the level of details provided in the current version does not convince the reviewers that the proposed approach is already efficient and can scale up. This could be shown by fair comparison with existing approaches (e.g., hard-coded error back-propagation implementation with a fixed graph) in runtime and accuracy.